# A diagnostic classifier for gene expression-based identification of early Lyme disease

Venice Servellita[1,7], Jerome Bouquet[1,7], Alison Rebman[2], Ting Yang[2], Erik Samayoa[1], Steve Miller[1], Mars Stone[3], Marion Lanteri[3], Michael Busch [3], Patrick Tang [4], Muhammad Morshed[5], Mark J. Soloski[2], John Aucott[2] & Charles Y. Chiu [1,6 ✉]

## Abstract

**Background** Lyme disease is a tick-borne illness that causes an estimated 476,000 infections annually in the United States. New diagnostic tests are urgently needed, as existing antibody-based assays lack sufficient sensitivity and specificity.

**Methods** Here we perform transcriptome profiling by RNA sequencing (RNA-Seq), targeted RNA-Seq, and/or machine learning-based classification of 263 peripheral blood mononuclear cell samples from 218 subjects, including 94 early Lyme disease patients, 48 uninfected control subjects, and 57 patients with other infections (influenza, bacteremia, or tuberculosis). Differentially expressed genes among the 25,278 in the reference database are selected based on ≥1.5-fold change, ≤0.05 $p$ value, and ≤0.001 false-discovery rate cutoffs. After gene selection using a k-nearest neighbor algorithm, the comparative performance of ten different classifier models is evaluated using machine learning.

**Results** We identify a 31-gene Lyme disease classifier (LDC) panel that can discriminate between early Lyme patients and controls, with 23 genes (74.2%) that have previously been described in association with clinical investigations of Lyme disease patients or in vitro cell culture and rodent studies of *Borrelia burgdorferi* infection. Evaluation of the LDC using an independent test set of samples from 63 subjects yields an overall sensitivity of 90.0%, specificity of 100%, and accuracy of 95.2%. The LDC test is positive in 85.7% of seronegative patients and found to persist for ≥3 weeks in 9 of 12 (75%) patients.

**Conclusions** These results highlight the potential clinical utility of a gene expression classifier for diagnosis of early Lyme disease, including in patients negative by conventional serologic testing.

## Plain language summary

Lyme disease is a bacterial infection spread by ticks and there are nearly half a million cases a year in the United States. However, the disease is difficult to diagnose and existing laboratory tests have limited accuracy. Here, we develop a new genetic test, described as a Lyme disease classifier (LDC), for diagnosing early Lyme disease from blood samples by assessing the patient's response to the infection. We find that the LDC can identify early Lyme disease patients (those presenting with symptoms within weeks of a tick bite) accurately, even before standard laboratory tests turn positive. In the future, the LDC may be clinically useful as a test for Lyme disease to diagnose patients earlier in the course of their illness, thus guiding more timely and effective treatment for the infection.

[1] Department of Laboratory Medicine, University of California, San Francisco, CA, USA. [2] Lyme Disease Research Center, Division of Rheumatology, Department of Medicine, Johns Hopkins School of Medicine, Baltimore, MD, USA. [3] Blood Systems Research Institute, San Francisco, CA, USA. [4] Sidra Medical and Research Center, Doha, Qatar. [5] British Columbia Centre for Disease Control, Vancouver, BC, Canada. [6] Department of Medicine, Division of Infectious Diseases, University of California, San Francisco, CA, USA. [7] These authors contributed equally: Venice Servellita, Jerome Bouquet. ✉email: charles.chiu@ucsf.edu

Lyme disease is a systemic tick-borne infection caused by *Borrelia burgdorferi* sensu lato and the most common vector-borne disease in the United States[1]. Lyme disease can cause arthritis, facial palsy, neuroborreliosis (neurological disease including meningitis, radiculopathy, and encephalitis), and even myocarditis resulting in sudden death[2]. Most patients treated with appropriate antibiotics recover rapidly and completely, but 5–15% of patients develop persistent or recurring symptoms. When prolonged and associated with functional disability, patients are considered to have post-treatment Lyme disease syndrome (PTLDS)[3,4]. The failure to diagnose and treat Lyme disease in a timely fashion results in higher morbidity and protracted recovery times[5].

Diagnosis of early Lyme disease is challenging[6]. Clinical manifestations can be highly variable, presenting as non-specific "flu-like" symptoms, and a characteristic bullseye erythema migrans (EM) rash is seen only 60–70% of the time[7]. Available FDA-approved serologic assays, including two-tier antibody testing recommended by the CDC for diagnosis, are negative in up to 40% of early Lyme patients[8–10]. Nucleic acid testing is hindered by low titers of *B. burgdorferi* in the blood during acute infection, with only 20–62% reported sensitivity of detection[11,12].

The advent of the genomics era has spurred the development of diagnostic tests based on transcriptome ("RNA-Seq") analyses of the human host response[13]. Classification by gene expression profiling has been useful in the identification of various infections, including *Staphylococcal* bacteremia[14], active versus latent tuberculosis[15], influenza[16,17], and COVID-19[18,19]. Transcriptome profiling of peripheral blood mononuclear cells (PBMCs)[20] or EM skin lesions[21] from patients with early Lyme disease has demonstrated pronounced inflammatory responses predominated by interferon signaling. Machine learning (ML)-based analyses of RNA-Seq data have been used for cancer classification[22], but to date have not yet been applied for infectious disease diagnosis. Here we sought to leverage iterative ML analyses of global and targeted RNA-Seq data to define a panel of differentially expressed genes (DEGs) to distinguish Lyme disease from non-Lyme controls. This panel, referred to as a Lyme disease classifier (LDC), consisted of 31 genes and was able to diagnose Lyme disease with >95% accuracy, including in >85% of Lyme seronegative patients.

## Methods

**Patient information.** Patient enrollment, chart review, collection of clinical samples, and analysis of clinical samples by transcriptomic profiling or targeted RNA sequencing were done under protocols approved by the Institutional Review Boards of Johns Hopkins University (JHU) (JHU IRB # NA_00011170) and the University of California, San Francisco (UCSF IRB # 17–241124211). Written informed consent was obtained from all JHU Lyme disease and uninfected control patients for enrollment into the study. No consents were obtained from other, non-JHU patients since only remnant clinical samples from these patients were used, and the samples were analyzed under protocols approved by the UCSF IRB as part of a "no subject contact" biobanking study with waiver of consent (UCSF IRB #17–2411).

All 94 Lyme disease subjects included in this study presented with a physician documented EM of ≥5 cm and either concurrent flu-like symptoms that included at least one of the following: fever, chills, fatigue, headache, and/or new muscle or joint pains or dissemination of the EM rash to multiple skin locations. Controls ($n = 26$) were enrolled from the same physician practice as cases. Two-tier serological Lyme disease testing was performed on clinical Lyme patients by a clinical reference laboratory (Quest Diagnostics) at the first visit and at 3 weeks, following a standard

3-week course of doxycycline treatment. Patients found to be Lyme seropositive at the first visit did not get repeat testing. Seropositivity was assessed according to established CDC criteria[23], including the requirement that patients have had symptoms for less than or equal to 30 days for Lyme diagnosis by positive ELISA and IgM testing. All controls were required to have a negative Lyme serologic test and no clinical history of Lyme disease to be enrolled in the study. All Lyme disease patients and controls were collected in Maryland, USA, an area highly endemic for Lyme disease.

PBMC samples from 57 patients diagnosed with other infections were collected at the UCSF, and 22 controls (asymptomatic blood donors) were collected at the Blood Systems Research Institute in San Francisco, California. Patients with other infections were diagnosed with either bacteremia ($n = 21$), caused by *Enterococcus faecium*, *Escherichia coli*, *Klebsiella pneumoniae*, *Staphylococcus aureus*, *Staphylococcus epidermidis*, or *Streptococcus pneumoniae* by standard plate culture, or influenza ($n = 36$) by positive RT-PCR testing (Luminex NxTAG Respiratory Pathogen Panel). PBMC samples from 19 adults, 9 patients diagnosed with tuberculosis using an interferon-gamma release assay (Oxford Immunotec T-SPOT.TB), and 10 uninfected controls, were collected at the British Columbia Centre for Disease Control in Vancouver, Canada.

PBMCs were isolated from freshly collected whole blood in EDTA tubes (kept at 4 °C for <24 h) using Ficoll (Ficoll-Paque Plus, GE Healthcare) and total RNA was extracted from $10^7$ PBMCs using TRIzol reagent (Life Technologies).

**Transcriptome sequencing.** Messenger RNA was isolated with the Oligotex mRNA mini kit (Qiagen). The Scriptseq RNA-Seq library preparation kit (Epicentre) was used to generate the RNA-Seq libraries according to the manufacturer's protocol. Libraries were sequenced as 100 bp paired-end reads on a HiSeq 2000 instrument (Illumina).

Samples were processed in two batches (Fig. 1). Set 1 corresponds to samples from 28 Lyme disease patients and 13 matched control samples as previously described[20]. Set 2 corresponds to samples from 13 new Lyme disease and 6 matched control samples prepared and sequenced alongside samples from 6 influenza and 6 bacteremia patients. One sample was not included in the pooled analysis due to insufficient read counts.

**Transcriptome RNA-Seq data analyses.** Paired-end reads were mapped to the human genome (hg19), followed by annotation of exons and calculation of FPKM (fragments per kilobase of exon per million fragments mapped) values for all 25,278 expressed genes with version 2 of the TopHat/Cufflinks pipeline[24]. Differential expression of genes was calculated using the variance modeling at the observational level transformation[25], which applies precision weights to the matrix count, followed by linear modeling with the Limma package. Genes were considered to be differentially expressed when the change was ≥1.5-fold, the $p$ value ≤ 0.05, and the adjusted $p$ value (or false-discovery rate, FDR) was ≤0.001[26].

**Targeted RNA sequencing.** Quantitative analysis of a custom panel of transcripts of interest was performed using a targeted RNA enrichment sequencing approach that incorporated an anchored multiplex PCR technique. PBMC samples (~1 million cells) were extracted using Zymo DirectZol RNA Miniprep Kit with on-column DNase following the manufacturer's instructions. Reverse transcription was performed using the Illumina TruSeq Targeted RNA Expression Kit on 50 ng of RNA according to the manufacturer's instructions. A custom panel of

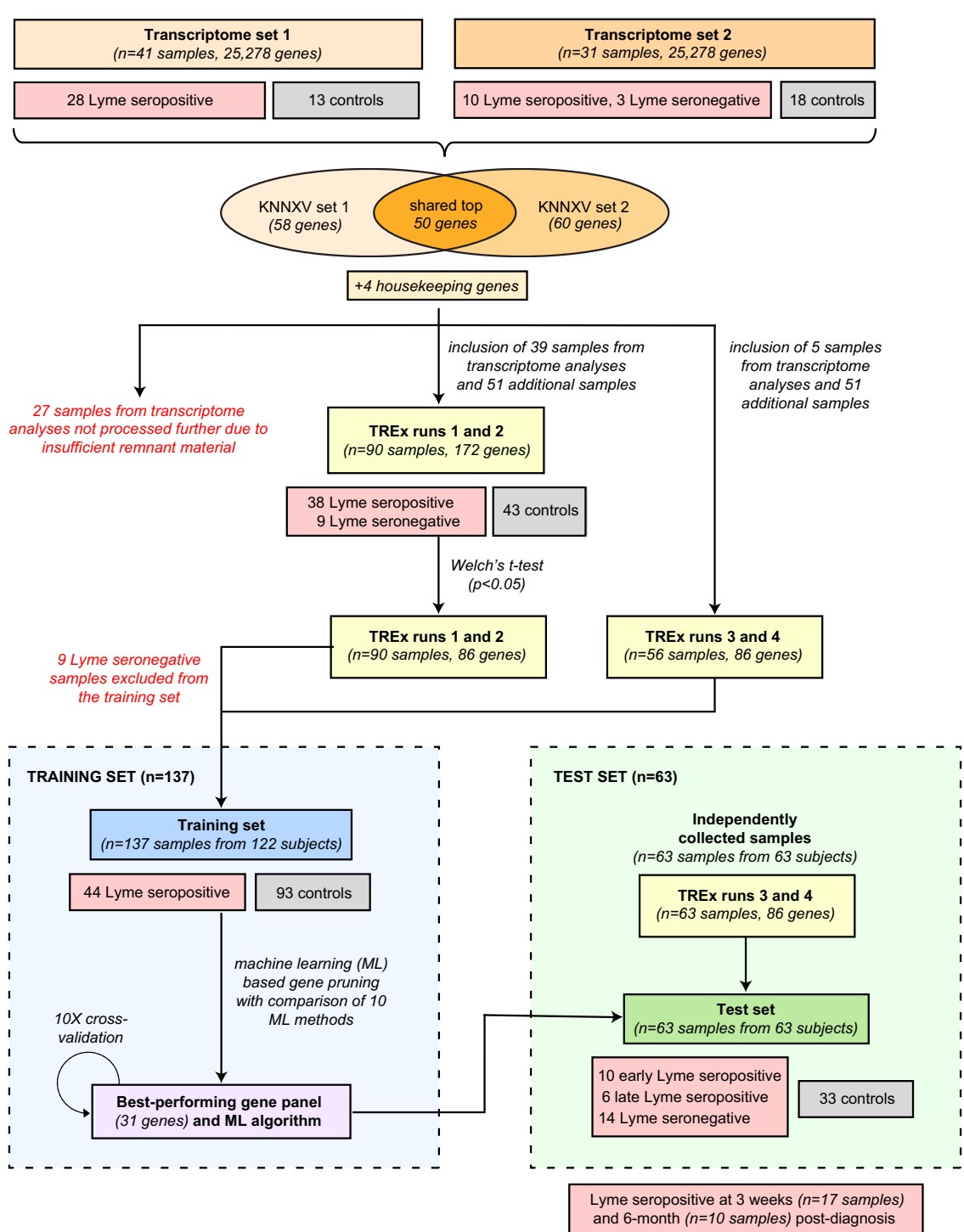

**Fig. 1 Flowchart of the approach used to develop and validate a 31-gene Lyme disease classifier panel for identification of early Lyme disease.** DEGs differentially expressed genes, KNNXV k-nearest neighbor cross-validation, TREx targeted RNA expression sequencing.

oligoucleotides representing the genes of interest was designed and ordered using the Illumina DesignStudio platform. This pool of oligonucleotides, each attached to a small RNA sequencing primer (smRNA) binding site, was used to hybridize, extend, and ligate the second strand of cDNA from targeted genes of interest. Thirty-five cycles of amplification were then performed using primers with a complementary smRNA sequence. The resulting libraries were sequenced on an Illumina MiSeq to a depth of ~2500 reads per sample per gene. Expression counts per sample per gene were calculated on the instrument using MiSeq reporter

targeted RNA workflow software (revision C). Briefly, following demultiplexing and FASTQ file generation, reads from each sample were normalized in R and then aligned locally against references corresponding to targeted regions of interest using a banded Smith–Waterman algorithm[27].

**Machine learning**. The k-nearest neighbor classification with leave-one-out cross-validation algorithm (KNNXV)[8], as implemented on Genepattern[28], was used on the set of DEGs identified by RNA-Seq-

based transcriptome profiling, using a k of 3, signal-to-noise ratio feature selection, Euclidean distance, and by iteratively decreasing the number of features until reaching maximum accuracy.

Class prediction performance using receiver-operating characteristic (ROC) metric on targeted RNA sequencing read count results was tested using the glmnet[29] and caret[30] packages in R for ten different ML methods at default parameters: classification and regression trees ("rpart" method), generalized linear models ("glmnet" method), linear discriminant analysis ("lda" method), k-nearest neighbor ("knn" method), random forest ("rf" method), eXtreme Gradient Boosting ("xgbTree" method), neural networks ("nnet" method), linear and radial support vector machine ("svmLinear" and "svmRadial" methods), and nearest shrunken centroid ("pam" method). Subsequent feature selection and fitting of the glmnet or generalized linear models were performed using 10-fold cross-validation with regularization using lasso (least absolute shrinkage and selection operator) penalty and lambda (λ) parameter. The value of lambda that provided the minimum mean cross-validated error was used to determine the optimal set of genes.

**Statistical methods**. The performance of the classifier was evaluated with the use of ROC curves, calculation of area under the curve (AUC)[31], and estimates of sensitivity, specificity, positive predictive value, and negative predictive value. A Mann–Whitney nonparametric test was used for the analysis of continuous variables, and Fisher's exact test was used for categorical variables. All confidence intervals were reported as two-sided binomial 95% confidence intervals. Statistical analysis was performed, and plots were generated using R software, version 4.0.3 (R Project for Statistical Computing).

**Reporting summary**. Further information on research design is available in the Nature Research Reporting Summary linked to this article.

## Results

The study comprised a total of 263 samples from 218 subjects (Table 1 and Supplementary Data 1). The 218 subjects included 94 Lyme disease patients, 66 infected "non-Lyme" controls with influenza ($n = 36$), tuberculosis ($n = 9$), and other bacteremia ($n = 21$), and 58 uninfected asymptomatic controls. All Lyme patients, including 61 seropositive and 33 seronegative by clinical two-tiered antibody testing, had documented EM rash and history of tick exposure at the time of presentation and were enrolled in the "Study of Lyme disease Immunology and Clinical Events" study at the Johns Hopkins Medical Institute. Control subjects categorized as uninfected asymptomatic were from regions with an incidence of Lyme disease of ≤0.2% (San Francisco, California and Vancouver, British Columbia) or had a negative Lyme serology test and no clinical history of tick-borne disease. No significant differences in age or sex were noted between Lyme and control subjects.

Transcriptome profiling using RNA-Seq was initially performed on PBMC samples from 72 subjects, including 41 Lyme patients and 31 controls (Fig. 1). Included were 41 samples from 28 Lyme patients and 13 uninfected controls (set 1), as previously reported[20]. For the remaining 31 samples from 13 Lyme patients and 18 controls (set 2), a mean of 30 (±17 standard deviation) million reads was generated per sample (Supplementary Fig. 1). No batch effect based on the geographic site of the collection was observed (Supplementary Fig. 2). DEGs were selected separately for each set of PBMC samples using the KNNXV ML feature selection algorithm[32]. The best accuracy for sets 1 and 2 was achieved using a panel of 58 and 60 genes, respectively.

**Table 1 Performance characteristics of the 31-gene Lyme disease classifier.**

| Study subjects | No. of samples tested | No. classified as Lyme (%) |
|---|---|---|
| *Training set*[a] | 137 | |
| Serologically confirmed Lyme disease[b] | 44 | 39 (89) |
| Seropositive at time of presentation | 26 | 23 (88) |
| Seropositive at 3 weeks | 18 | 16 (89) |
| Controls | 93 | 12 (13) |
| Uninfected | 57 | 9 (16) |
| Bacteremia | 9 | 2 (22) |
| Influenza | 21 | 1 (5) |
| Tuberculosis | 6 | 0 (0) |
| *Test set*[c] | 63 | |
| Serologically confirmed Lyme disease[b] | 16 | 15 (94) |
| Seropositive at time of presentation (early seroconversion) | 10 | 10 (100) |
| Seropositive at 3 weeks (late seroconversion) | 6 | 5 (83) |
| Seronegative Lyme disease | 14 | 12 (86) |
| Controls | 37 | 0 (0) |
| Uninfected | 15 | 0 (0) |
| Bacteremia | 6 | 0 (0) |
| Influenza | 9 | 0 (0) |
| Tuberculosis | 3 | 0 (0) |
| *Longitudinally collected samples* | | |
| Lyme disease 0 week | 16 | 14 (88) |
| Lyme disease 3 weeks post diagnosis | 17 | 13 (76) |
| Lyme disease 6 months post diagnosis | 10 | 3 (30) |

[a]Sensitivity 95.5% [84.1–100%], specificity 86.0% (77.4–98.98%), accuracy 87.6% (80.9–92.6%), area under the curve (AUC) 97.2% (95.0–99.3%).
[b]Positive by two-tiered Lyme antibody testing.
[c]Sensitivity 90.0% (83.3–100%), specificity 100% (90.0–100%), accuracy 95.2% (86.7–99.0%), AUC 98.2% (95.7–100%).

These genes, along with an additional top 50 DEGs that were ranked according to adjusted *p* value/FDR in order of decreasing significance and did not overlap with the two panels, were then combined into a 172-gene targeted RNA sequencing panel (Supplementary Data 2). The 172-gene panel was used to test 90 samples (38 Lyme seropositive, 9 Lyme seronegative, and 43 controls) over 2 targeted RNA expression sequencing runs (TREx, "targeted RNA expression" runs 1 and 2). A subset of 86 genes out of 172 (50%) with the maximum differences in gene expression between Lyme and "non-Lyme" control samples across the first 2 TREx runs was identified using Welch's *t*-test at a $p < 0.05$ cutoff. The smaller 86-gene panel was then used to analyze an additional 119 samples in TREx runs 3 and 4.

Next, ML-based methods were applied to select from the list of 86 candidate genes and determine the optimal combination of genes and classification model for the LDC. We randomly partitioned samples from TREx runs 1–4 into a training set or test set. After ensuring that the training set consisted entirely of samples from laboratory-confirmed ("Lyme seropositive") Lyme disease patients and that no prior analyses had been performed on the independent test set, 137 and 63 samples were assigned to the training and test sets, respectively, at an approximately 2:1 (68.5%:31.5%) ratio. The training set was used to evaluate ten different ML algorithms for feature and model selection while varying the number of features (genes) from 1 to 86 for

discriminating Lyme from non-Lyme patients using a 10-fold cross-validation scheme (Supplementary Fig. 3). A generalized linear model ("glmnet") was found to provide the highest AUC-ROC statistic (97.2%) with the AUC-ROC of other methods varying from 70 to 93%. The optimal cutoff as determined by Youden's J statistic (Youden, 1950) was 0.3. The highest AUC and lowest rate of misclassification error were found with a panel of 31 genes (Fig. 2A).

Based on the expression of the 31 genes in the finalized LDC panel, a disease score ranging from 0 to 1 was calculated, with a score >0.3 classified as Lyme and <0.3 as "non-Lyme". Compared to two-tier Lyme antibody testing as a reference gold standard, training set sensitivity, specificity, and AUC-ROC using this scoring metric were 95.5% (95% CI 84.1–100%), 86.0% (95% CI 77.4–98.9%), and 97.2 (95% CI 95.0–99.3%), respectively (Fig. 2B and Table 1). Five of 44 (11.4%) Lyme samples and 12 of 93 controls (12.9%) in the training set were misclassified (Fig. 2C). LDC results between subjects who were seropositive at presentation had comparable sensitivity to those who were seropositive after 3 weeks (Table 1, 88% versus 89%, respectively).

For the independent test set of 63 samples, the LDC classifier had an overall accuracy of 95.2% (95% CI 86.7–99.0%), with a sensitivity of 90% (95% CI 83.3–100%) and specificity of 100% (95% CI 90.9–100%) relative to two-tier Lyme antibody testing and based on misclassification of 1 Lyme seropositive and 2 Lyme seronegative samples (Fig. 2D, E). LDC results between subjects seropositive at presentation had higher sensitivity than those who were seropositive after 3 weeks (Table 1, 100% versus 83%, respectively). LDC sensitivities for Lyme seropositive and seronegative samples were 93.7% and 85.7%, respectively (Table 1).

The 31 identified genes on the panel were related to immune cell signaling ($n = 7$), cell division ($n = 6$), apoptosis ($n = 3$), cell growth and differentiation ($n = 3$), cell trafficking ($n = 2$), *B. burgdorferi* receptor-binding ($n = 2$), and 8 other functions ($n = 8$) (Fig. 2F). Many genes (23 of 31, 74.2%) had previously been described in association with cell culture ($n = 20$), murine ($n = 2$), and Lyme disease patient studies ($n = 3$) of *B. burgdorferi* infection (Supplementary Data 3).

To evaluate for the persistence of the LDC gene signature, we analyzed available serially collected samples from a subset of 18 clinical Lyme patients at 0 week (time of initial clinical presentation with EM rash) and 3 weeks (following completion of a 3-week course of doxycycline treatment) (Fig. 3). Among four Lyme seronegative cases, three (75%) had a discordant result, with negative Lyme serology but a positive LDC score of >0.3 (Fig. 3, P2–P4). Two of these three cases seroconverted at 3 weeks by IgM testing (Fig. 3, P2 and P4) but did not formally fulfill CDC criteria since the duration of illness from onset of symptoms was >30 days (although would be considered seropositive using a 6-week cutoff as suggested by others)[33], while the remaining seronegative/LDC-positive patient (Fig. 3, P3) was ELISA positive and had one and two bands for IgM and IgG, respectively, at 3 weeks, appeared close to seroconverting. Among the 4 cases with late seroconversion 3 weeks after the presentation (Fig. 3, P5–P8), 3 of 4 (Fig. 3A, P6–P8) were positive by LDC testing at time 0 week, while P5 was negative at 0 week but positive at 3 weeks. Ten of 13 cases (76.9%) that were LDC positive at time 0 remained persistently positive at 3 weeks (Fig. 3, P2, P7, P8, P9, P10, P11, P15, P16, P17, and P18), while the remaining 3 (Fig. 3, P6, P12, and P14) showed a decline in the LDC score below the 0.3 threshold.

Samples from ten patients collected at 3 weeks and/or 6 months after the clinical presentation of Lyme disease were available and, based on LDC testing, could be assigned into two subgroups with different longitudinal trajectories (Fig. 4). One subgroup (Fig. 4, I) contained three patients with positive LDC scores at 0 week (Fig. 4, P2, P12, and P14) that declined at 3 weeks but rebounded by 6 months. P12 and P14 had persistent symptoms at 6 and 12 months, respectively, but without the functional disability to meet clinical criteria for PTLDS[3,4]. The other subgroup (Fig. 4, II) contained seven patients who had gradual declines in LDC score from 0 week to 6 months. Among these seven patients, two were symptomatic at 6 months but returned to usual state of health at 1 year (Fig. 4, P13 and P16), while one Lyme seronegative patient diagnosed with clinical PTLDS was negative by LDC testing at all three time points (Fig. 4, P1).

Unfortunately, 6-month samples were not available for two Lyme disease patients who met clinical criteria for PTLDS and had a persistently positive LDC signature at 3 weeks (Fig. 3B, P4 and P9).

## Discussion

Here we applied transcriptome profiling, targeted RNA-Seq, and iterative ML-based analyses to construct a 31-gene LDC with 90% sensitivity and 100% specificity in identifying clinical Lyme patients at the time of initial presentation. A condensed diagnostic panel of 31 multiplexed gene targets makes it amenable to implementation on commercial multiplexed nucleic acid testing instruments[34] or on targeted RNA next-generation sequencing platforms, with the latter being used in 2020–2021 for clinical SARS coronavirus 2 (SARS-CoV-2) testing under FDA Emergency Use Authorization[35]. We also found that 77% of Lyme disease patients with a positive LDC at initial presentation remained positive for at least 3 weeks, consistent with earlier work on the Lyme disease transcriptome[20]. This observation indicates that an LDC classifier may be useful for Lyme disease diagnosis during the approximately 3-week "window period" prior to the generation of detectable antibody levels by two-tiered testing[23]. Taken together, the LDC classifier meets four of the five characteristics of an "ideal" Lyme disease diagnostic, as described by Schutzer et al.[8], including high sensitivity in early infection, high specificity, ≤24 h turnaround time (if implemented on a multiplexed nucleic acid testing platform), and testing from easily collected samples such as blood. Thus, the LDC classifier may be useful as a complementary diagnostic to serologic testing, which exhibits high sensitivity (95–100%) in later stages of Lyme disease (the sole remaining characteristic out of 5), but inadequate sensitivity (29–77%) in early Lyme[10,36].

As expected, most of the genes (74%, 23 of 31) in the LDC classifier panel had previously been reported as related to Lyme disease based on in vitro and in vivo investigations. However, the LDC would have been near impossible to construct a priori given that selection of an optimal set of genes would have been difficult and that 8 of the 31 (25.8%) genes had not been previously described in the literature. Notably, only 7 (22.5%) genes in the panel were associated with immune cell signaling, of which 3 (9.7%) were related to interferon signaling, in contrast with prior reports demonstrating strong immune and inflammatory responses in early Lyme disease[20,21,37,38]. Unlike these previous studies, here we incorporated controls from patients with acute febrile infections from viruses (influenza) or other bacteria, potentially explaining why only a minority of LDC genes were associated with immune cell signaling. Instead, many of the identified genes in the LDC were related to cell division and proliferation, autophagy, and apoptosis. It has previously been shown that PBMCs from patients with Lyme disease exhibit proliferation in vitro to *B. burgdorferi* infection[39]. *B. burgdorferi* has also been shown to induce autophagy in infected PBMCs resulting in the production of cytokines such as interleukin-1β[40]. In addition, phagocytosis of *B. burgdorferi* induces apoptosis in

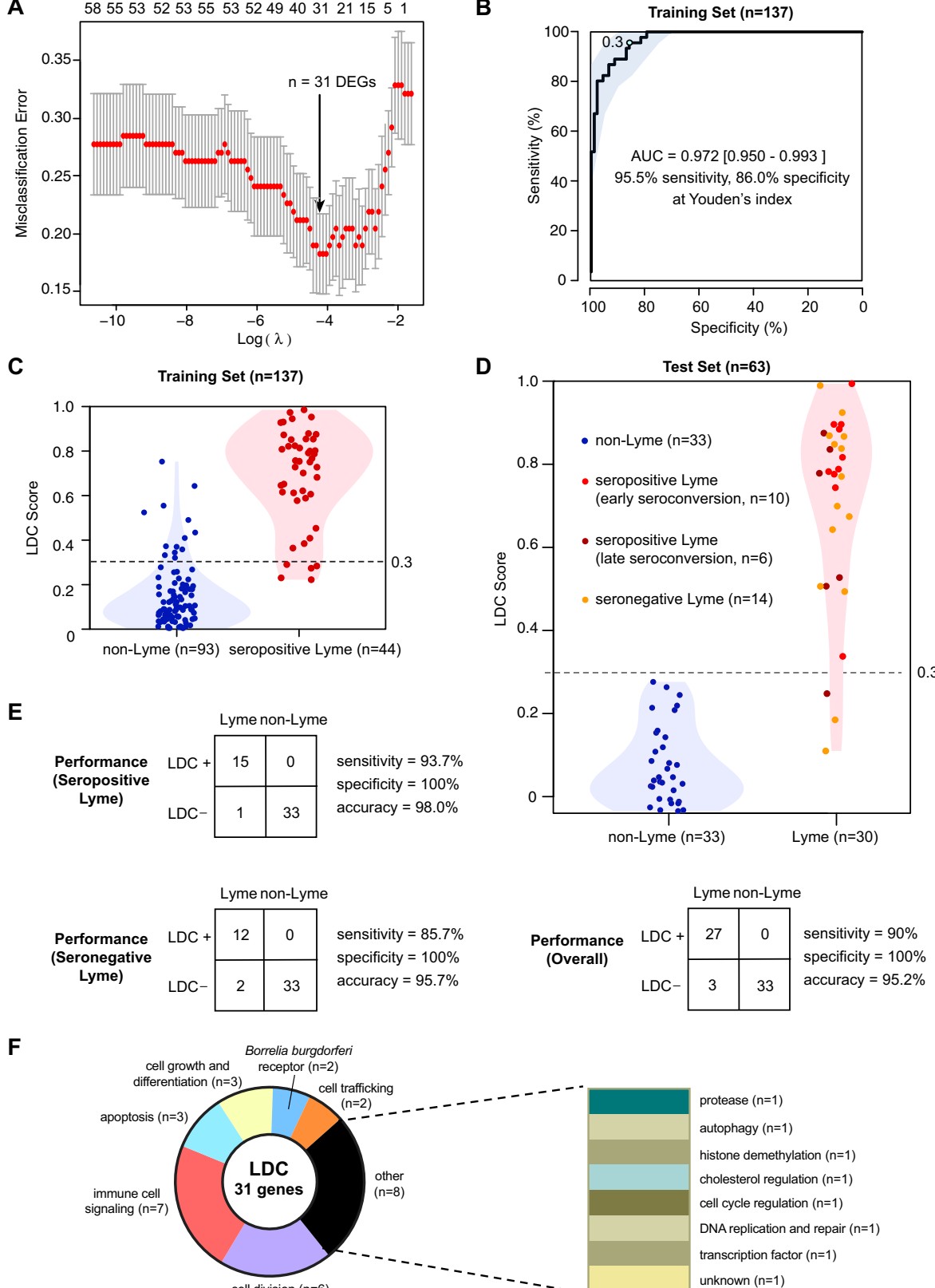

**Fig. 2 A 31-gene Lyme disease classifier derived using the generalized linear model machine learning algorithm. A** Chart of misclassification error versus number of genes considered and related log(lambda) statistic. **B** Receiver-operating characteristic (ROC) curve of the performance of the LDC using a training set of 44 Lyme seropositive and 93 "non-Lyme" control samples. The cutoff for positivity according to Youden's J statistic is 0.3. **C** Violin plots of the LDC score for an independent test set of 63 samples. **D** Violin plots of the LDC score for the training set of 137 samples. **E** 2 × 2 contingency tables of LDC test set performance overall and for seropositive (serologically confirmed) and seronegative Lyme cases. **F** Pie chart of signaling pathways associated with the 31 genes comprising the LDC panel.

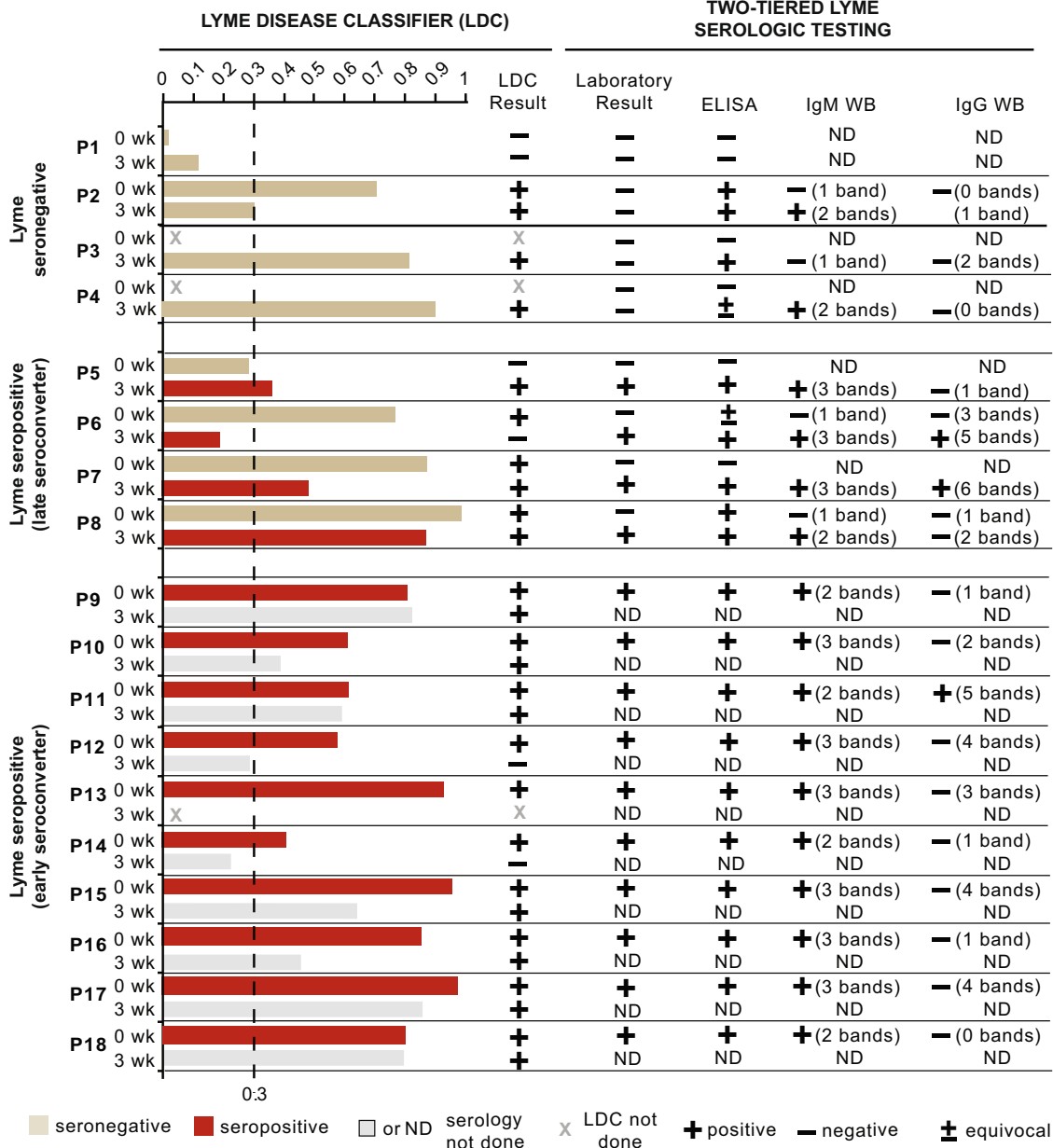

**Fig. 3 Longitudinal testing of clinical Lyme patients using the Lyme disease classifier.** A comparison between the LDC score and results from two-tiered Lyme serologic testing for Lyme seronegative and Lyme seropositive (both early and late seroconversion) patients at 0 and 3 weeks. Patients testing Lyme seropositive at 0 week did not get repeat serologic testing. CDC criteria for a positive Lyme serology include a positive screening ELISA and either ≥2 of 3 bands on reflex IgM testing (in patients with signs and symptoms lasting <30 days) or ≥5 of 10 bands on reflex IgG testing[23]. LDC Lyme disease classifier, ELISA enzyme-linked immunosorbent assay, WB western blot, IgM immunoglobulin M, IgG immunoglobulin G.

human monocytes[41] and also in neuronal cells of the dorsal root ganglia[42]. Genes associated with these signaling pathways may be more specific to Lyme disease and thus more useful as diagnostic biomarkers than those focused solely on immune and inflammatory responses. Further research on the genes identified in the LDC classifier to investigate their involvement in *Borrelia* pathogenesis is warranted in future studies.

Prior studies have used gene expression to profile Lyme disease patients from PBMCs[20,37,38], although our study incorporates larger numbers of Lyme disease cases and controls. The three previously reported studies present similar findings showing an increase in immune and inflammatory response genes, particularly those interferon-regulated, in Lyme disease cases relative to uninfected controls. The study by Clarke, et al.[37] also reported the

development of a diagnostic classifier of 20 genes for early Lyme disease, but the performance was not evaluated with an independent test set. The study by Petzke, et al.[38] reported two kinds of classifiers for discriminating between Lyme disease cases and controls and between Lyme disease cases that resolve after treatment and those that progress to having persistent symptoms. All these classifiers are limited by the absence of controls from other viral and bacterial infections to exclude overlapping immune and inflammatory response genes. In fact, only two genes in our LDC classifier, TYMS, a DNA replication and repair gene, and GRN, a cell proliferation gene, are shared with these prior classifiers[37,38]. Other "omics" technologies have been used to develop classifiers for Lyme disease. For example, a previous study reported a metabolomic signature with 88% sensitivity and

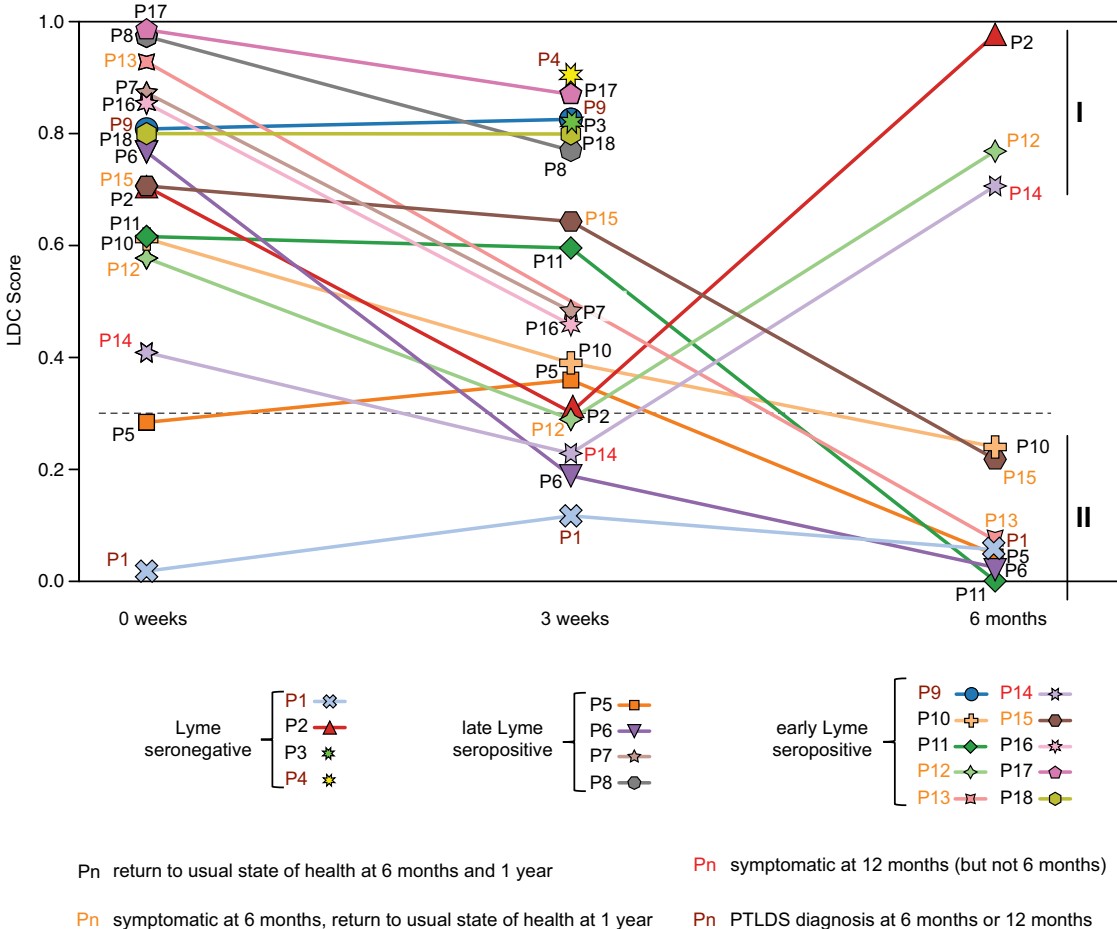

**Fig. 4 Lyme disease classifier scores from longitudinally collected patient samples.** Plots of the LDC score in 18 Lyme disease patients from available samples collected at 0 week, 3 weeks, and 6 months. An LDC result is considered positive if the LDC score is above the 0.3 cutoff as determined from training set data using Youden's J statistic. Patients are labeled P1–P18.

95% specificity for the identification of seropositive Lyme[43], although the controls in that study were different (infectious mononucleosis, fibromyalgia, severe periodontitis, and syphilis).

One limitation of the current study is the absence of controls from other, less common tick-borne (e.g., babesiosis, anaplasmosis, ehrlichiosis, rickettsiosis, and Powassan virus infection) and spirochetal (e.g., syphilis, leptospirosis) infections. However, nearly all of these other tick-borne and spirochetal infections can be diagnosed by conventional microbiological molecular and/or serologic testing[44]. In addition, we previously reported more overlap in the transcriptomic signature of Lyme disease with viral (influenza) infection than with bacterial infection[20]. This suggests that the human host response to Lyme disease is likely different from other tick-borne and spirochetal infections. The finding of 23 of 31 genes in the classifier being related to *Borrelia* infection also supports the contention that the LDC is specific to Lyme disease. Another limitation is the small size of longitudinally collected samples at 3 weeks ($n = 17$) and 6 months ($n = 10$). Here we focused on a classifier for early Lyme disease based on host gene expression. Further investigation will be needed to investigate its potential role in the evaluation of Lyme disease patients with chronic symptoms and/or PTLDS. Finally, it can be challenging to develop and clinically validate an RNA expression-based assay for 31 genes simultaneously, However, it may be feasible to decrease the number of genes on the panel without unduly sacrificing performance (Fig. 2A), and FDA authorization

of targeted omics-based tests for COVID-19[35] suggests a potential regulatory pathway for the deployment of a multiplexed Lyme diagnostic in the near future.

As ~86% of samples from patients persistently seronegative at 0 and 3 weeks were correctly classified as Lyme, our LDC classifier may allow more accurate stratification of presumptive Lyme patients testing negative by serology. In the absence of "gold-standard" testing, it cannot be proven that these seronegative patients were infected by *B. burgdorferi*. Nevertheless, documentation of EM rash in all Lyme patients in this study, even in those who tested seronegative, concurrent "flu-like" symptoms, and enrollment during tick season in a region highly endemic for Lyme disease suggest that this may indeed be the case. Evidence in support of infection is also provided by the finding that three of the four LDC-positive, seronegative patients exhibited borderline serologic responses just outside of formal CDC criteria for seropositivity. Conversely, the remaining seronegative Lyme patient, who was also negative by LDC testing (Figs. 3 and 4, P1), appears to be a likely *bona fide* Lyme-negative case, despite being incidentally diagnosed with PTLDS. More accurate discrimination of Lyme patients using the LDC may be clinically useful by prompting diagnostic workup for a different tick-borne disease or other acute illness. The identification of a subgroup of three patients (out of ten) with a persistently positive LDC signature at 6 months, two of whom had ≥6 months of persistent symptoms, warrants further study on the potential utility of the LDC for

diagnosis and monitoring of Lyme disease patients with chronic symptoms.

## Data availability

All data in this study were submitted to the National Institutes of Health (NIH) database of Genotypes and Phenotypes (dbGaP) (read count tables, raw FASTQ files for transcriptome sets 1 and 2 accession number phs002794.v1.p1). Public summary phenotype data are available at the dbGaP study report web page: https://www.ncbi.nlm.nih.gov/projects/gap/cgi-bin/study.cgi?study_id=phs002793.v1.p1. Individual-level data, including transcriptomic sequencing data, are available for download by authorized investigators via https://view.ncbi.nlm.nih.gov/dbgap-controlled. The sequencing data are only available via restricted access as patients did not consent for the public release of their data and to protect patient confidentiality. Metadata for the 263 clinical samples included in this study are provided in Supplementary Data 1. Source data used to generate the main figures are provided in Supplementary Data 4.

## Code availability

Code used to reproduce the ML analysis for LDC model prediction and feature selection has been deposited in a Zenodo repository (doi: 10.5281/zenodo.5987532)[45].

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

## Acknowledgements

This work was supported by grants from the Bay Area Lyme Foundation, the Steven and Alexandra Cohen Foundation, the Benioff Foundation, the Swartz Foundation, the Stabler Foundation, the Global Lyme Alliance, and the National Institutes of Health (grants R01-HL105704 and P30-AR05350), We would like to thank Yvonne Simpson for identifying and preparing tuberculosis patient and control samples for this study.

## Author contributions

J.B. and C.Y.C. conceived of and designed the study. J.B. performed the experiments. V.S., J.B., A.R., T.Y., E.S., S.M., M.S., M.L., M.B., P.T., M.M., M.J.S., and J.A. collected samples and associated clinical and laboratory metadata. V.S., J.B., A.R., T.Y., and C.Y.C. analyzed clinical and epidemiological data. V.S., J.B., and C.Y.C. analyzed the gene expression data. V.S., J.B., and C.Y.C. wrote the manuscript. V.S. and C.Y.C. designed the figures. V.S., J.B., M.J.S., J.A., and C.Y.C. edited the manuscript.

## Competing interests

C.Y.C. and J.A. are on the scientific advisory board for the Bay Area Lyme Foundation. The other authors declare no competing interests.

## Additional information

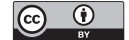 ns license, unless indicated otherwise in a credit line to the material. If material is not included in the article's Creative Commons license and your intended use is not permitted by statutory regulation or exceeds the permitted use, you will need to obtain permission directly from the copyright holder. To view a copy of this license, visit http://creativecommons.org/licenses/by/4.0/.

© The Author(s) 2022

