## [Peer Review File · Communications Medicine]

Reviewers' comments:

Reviewer #1 (Remarks to the Author):

Overview: this is a report focused on the utilization of transcriptomic analysis and machine learning to develop a characteristic profile for Lyme disease—a Lyme disease classifier (LDC). The researchers used a sample set of 263 samples, including 94 Lyme and multiple controls. A training set was used to determine the best algorithm, which encompassed 31 genes by a generalized linear model. The process is clearly presented in Figure 1. The targeted RNA expression analysis (Trex) was found to be highly sensitive and specific in the early stage of LD (when either seropositive or seronegative from 3 weeks after diagnosis). Its efficacy in later stage may not be so high, but the authors point out that the test meets key criteria needed to improve early Lyme diagnosis.

This is a well-written, controlled and statistically analyzed report. Only minor concerns exist.

- The paper would benefit from a direct comparison between their LDC results and two-tier serology results, predominantly in the early stage. This could be added in the text of results or in Table 1.
- Table 1 needs to be reformatted--the test set data are cut off.
- In the discussion, when describing the limitations, the authors should include a bit about the feasibility of this assay for routine diagnostic testing. It doesn't seem that it can be reduced to a few biomarkers, but that all 31 genes need to be analyzed for relative transcription.

Reviewer #2 (Remarks to the Author):

The authors identified a 31-gene Lyme disease classifier (LDC) to discriminate early Lyme patients from “non-Lyme” controls infected with influenza, bacteremia, or tuberculosis or uninfected asymptomatic controls, using transcriptome profiling by RNA-Seq, targeted RNA sequencing, and machine learning (ML)-based classification of 218 subjects. Evaluation of the LDC using an independent test set of samples from 63 subjects (16 Lyme seropositive patients, 14 Lyme seronegative patients, and 33 controls) yielded an overall sensitivity of 90.0%, specificity of 100%, and accuracy of 95.2%. There are several major problems:

1. Have the authors uploaded the data onto GEO? There should be an accession number. The authors should give a temporary access code for the reviewers to check the data.
2. The authors should store the scripts of the prediction model on GitHub or other publicly available websites.
3. The gene selection procedure was not clear: “Differentially expressed genes (DEGs) were selected separately for each set of PBMC samples using the k-nearest neighbor classification with leave-one-out cross-validation (KNNXV) ML algorithm. The best accuracy for sets 1 and 2 was achieved using a panel of 58 and 60 genes, respectively. These genes, along with an additional top 50 DEGs shared between the two sets that did not overlap with the 2 panels and 4 housekeeping genes, were then combined into a 172-gene targeted RNA sequencing panel”
4. What did you mean top? Did the authors rank the gene list? If so, how did they rank the genes?
5. The authors should do feature selection rather than just use DEG for prediction model construction. Numerous studies have shown that DEGs were not good features and feature selection can identify better gene signatures (PMID: 34305880, 33292121, 32411685).
6. The authors should do a more in-depth biological function analysis of the selected gene

signatures.

Reviewers' comments:

Reviewer #1 (Remarks to the Author):

Overview: this is a report focused on the utilization of transcriptomic analysis and machine learning to develop a characteristic profile for Lyme disease-a Lyme disease classifier (LDC). The researchers used a sample set of 263 samples, including 94 Lyme and multiple controls. A training set was used to determine the best algorithm, which encompassed 31 genes by a generalized linear model. The process is clearly presented in Figure 1. The targeted RNA expression analysis (Trex) was found to be highly sensitive and specific in the early stage of LD (when either seropositive or seronegative from 3 weeks after diagnosis). Its efficacy in later stage may not be so high, but the authors point out that the test meets key criteria needed to improve early Lyme diagnosis.

This is a well-written, controlled and statistically analyzed report. Only minor concerns exist.

We thank the reviewer for the positive comments regarding our manuscript.

-The paper would benefit from a direct comparison between their LDC results and two-tier serology results, predominantly in the early stage. This could be added in the text of results or in Table 1.

We would like to clarify that Table 1 already provides a summary of the comparison between the LDC and serology results (e.g. "serologically confirmed"). We clarify these comparisons in a revised Table 1 by addition of footnotes and added a description of the comparisons made and use of two-tier Lyme testing as a reference gold standard to the text of the results as suggested by the reviewer. Finally, we added the raw two-tier serology results at all available timepoints as columns to Supplementary Table 1.

-Table 1 needs to be reformatted--the test set data are cut off.

Table 1 has been reformatted and some of the extraneous columns regarding sensitivity, specificity, accuracy, and AUC moved to the footnotes.

-In the discussion, when describing the limitations, the authors should include a bit about the feasibility of this assay for routine diagnostic testing. it doesn't seem that it can be reduced to a few biomarkers, but that all 31 genes need to be analyzed for relative transcription.

Figure 2A suggests that fewer than 31 genes (n=15-31) would have similar performance. We make a note of this in the discussion and add a few sentences regarding the feasibility of the assay for routine diagnostic testing to the discussion in the limitations section as suggested the reviewer.

Reviewer #2 (Remarks to the Author):

The authors identified a 31-gene Lyme disease classifier (LDC) to discriminate early Lyme patients from “non-Lyme” controls infected with influenza, bacteremia, or tuberculosis or uninfected asymptomatic controls, using transcriptome profiling by RNA-Seq, targeted RNA sequencing, and machine learning (ML)-based classification of 218 subjects. Evaluation of the LDC using an independent test set of samples from 63 subjects (16 Lyme seropositive patients, 14 Lyme seronegative patients, and 33 controls) yielded an overall sensitivity of 90.0%, specificity of 100%, and accuracy of 95.2%. There are several major problems:

1. Have the authors uploaded the data onto GEO? There should be an accession number. The authors should give a temporary access code for the reviewers to check the data.

We are unable to upload the data to GEO because the Johns Hopkins University IRB has insisted that all data from Lyme disease patients and controls from JHU, including raw FASTQ transcriptome files, should be submitted into the restricted access dbGaP database. We have now submitted all read count tables and clinical metadata dbGaP under accession number phs002794.v1.p1. The data needs to be reviewed by an NIH dbGaP curator before we can submit the FASTQ files and make them available under restricted access, a process which can take 4-6 weeks (<https://www.ncbi.nlm.nih.gov/sra/docs/sra-dbgap-download-old/>).

In the meantime, we have arranged to make the read count table and raw FASTQ transcriptome files available to reviewers...[redacted]

2. The authors should store the scripts of the prediction model on GitHub or other publicly available websites.

Source data and code used to reproduce the ML analysis for LDC model prediction and feature selection have been deposited in a Zenodo repository (doi: 10.5281/zenodo.5987532), as suggested by the reviewer. It has been made public and is accessible here:

<https://zenodo.org/record/5987532#.YqIj5vXMJMw>

3. The gene selection procedure was not clear: “Differentially expressed genes (DEGs) were selected separately for each set of PBMC samples using the k-nearest neighbor classification with leave-one-out cross-validation (KNNXV) ML algorithm. The best accuracy for sets 1 and 2 was achieved using a panel of 58 and 60 genes, respectively. These genes, along with an additional top 50 DEGs shared between the two sets that did not overlap with the 2 panels and 4 housekeeping genes, were then combined into a 172-gene targeted RNA sequencing panel”

4. What did you mean top? Did the authors rank the gene list? If so, how did they rank the genes?

Genes were ranked according to adjusted p-value / FDR (false discovery rate). This is clarified in the revised text.

5. The authors should do feature selection rather than just use DEG for prediction model construction. Numerous studies have shown that DEGs were not good features and feature selection can identify better gene signatures (PMID: 34305880, 33292121, 32411685).

We would like to clarify that we did use feature selection predominantly for prediction model construction. KNNXV feature selection and the top 50 DEGs by adjusted p-value / FDR were used to create the first 172-gene RNA expression panel. The panel was then narrowed down to 31 genes using glmnet feature selection. We clarified in the revised text that KNNXV and models such as glmnet are feature selection methods.

6. The authors should do a more in-depth biological function analysis of the selected gene signatures.

The selected genes were analyzed for function in Figure 2E. This was incorrectly cited in the text of the results as 2D, which may have caused confusion and has been corrected in the revised text. We also cite and literature and review the function of the 31 genes in the final panel in Supplementary Table 3. We also add a sentence in discussion that further research on the genes identified in the LDC

classifier to investigate their involvement in Borrelia pathogenesis is warranted in future studies.

REVIEWERS' COMMENTS:

Reviewer #1 (Remarks to the Author):

All of my concerns have been addressed

Reviewer #2 (Remarks to the Author):

The authors have answered my questions.